# Encapsulation of Hydrophobic and Low-Soluble Polyphenols into Nanoliposomes by pH-Driven Method: Naringenin and Naringin as Model Compounds

**DOI:** 10.3390/foods10050963

**Published:** 2021-04-28

**Authors:** Mianhong Chen, Ruyi Li, Yuanyuan Gao, Yeyu Zheng, Liangkun Liao, Yupo Cao, Jihua Li, Wei Zhou

**Affiliations:** 1Key Laboratory of Tropical Crop Products Processing of Ministry of Agriculture and Rural Affairs, Agricultural Products Processing Research Institute, Chinese Academy of Tropical Agricultural Sciences, Zhanjiang 524001, China; mianhong_chen@163.com (M.C.); fannyloveme@163.com (Y.G.); yeyu_zheng@163.com (Y.Z.); liaoliangkunv@163.com (L.L.); yupo53@163.com (Y.C.); foodpaper@126.com (J.L.); 2State Key Laboratory of Food Science and Technology, Nanchang University, Nanchang 330047, China; 3Hainan Key Laboratory of Storage and Processing of Fruits and Vegetables, Zhanjiang 524001, China

**Keywords:** naringenin, naringin, nanoliposomes, pH-driven, stability

## Abstract

Naringenin and naringin are a class of hydrophobic polyphenol compounds and both have several biological activities containing antioxidant, anti-inflammatory and anti-tumor properties. Nevertheless, they have low water solubility and bioavailability, which limits their biological activity. In this study, an easy pH-driven method was applied to load naringenin or naringin into nanoliposomes based on the gradual reduction in their water solubility after the pH changed to acidity. Thus, the naringenin or naringin can be embedded into the hydrophobic region within nanoliposomes from the aqueous phase. A series of naringenin/naringin-loaded nanoliposomes with different pH values, lecithin contents and feeding naringenin/naringin concentrations were prepared by microfluidization and a pH-driven method. The naringin-loaded nanoliposome contained some free naringin due to its higher water solubility at lower pH values and had a relatively low encapsulation efficiency. However, the naringenin-loaded nanoliposomes were predominantly nanometric (44.95–104.4 nm), negatively charged (−14.1 to −19.3 mV) and exhibited relatively high encapsulation efficiency (EE = 95.34% for 0.75 mg/mL naringenin within 1% *w/v* lecithin). Additionally, the naringenin-loaded nanoliposomes still maintained good stability during 31 days of storage at 4 °C. This study may help to develop novel food-grade colloidal delivery systems and apply them to introducing naringenin or other lipophilic polyphenols into foods, supplements or drugs.

## 1. Introduction

Naringenin (5,7,4′-trihydroxyflavanone) and naringin (5,7,4′-trihydroxyflavone 7-rhamno-glucoside) are flavones abundantly existent in fruits, including grapefruit, tomato skin and oranges [1]. Naringin can be formed by naringenin at the position of the 7th carbon atom with neohesperidose (Figure 1), which promotes its solubility in water. Their health-promoting features have been explored in vivo and in vitro models, such as antioxidant activity [2,3], anti-inflammatory activity [4,5], anticancer activity [6,7], anti-diabetic activity [8] and protective effects on the central nervous system [9] and the cardiovascular system [10]. However, naringenin and naringin are hydrophobic and low-soluble polyphenols with low water solubility of 38 and 500 µg/mL at room temperature, respectively [11,12], and taste bitter, which greatly limits their application in clinical and functional foods. To overcome these limits, the solubility and bioavailability of naringenin and naringin have been improved by encapsulation in various kinds of carefully designed colloidal delivery systems, including cyclodextrins complexation [12], nanovehicles [13,14], liposomal [15] and ternary complex particles [16].

The pH-driven method is a new method for constructing a delivery system, which has received extensive attention due to its easy-to-handle, cost-saving, energy-saving and organic solvent-free properties during operation. The process of this method includes dissolving the bioactivator under strongly alkaline conditions and then adjusting the pH to neutral or acidic after blending with the carrier system. Hydrophobic polyphenols like curcumin have poor solubility in neutral acidity, while they can be dissolved due to deprotonation at alkaline pH. According to this property of some polyphenols, Pan et al. firstly loaded curcumin into casein nanoparticles by the pH-driven method [17]. Subsequently, a series of studies on the colloidal delivery systems based on the pH-driven method began to emerge, especially for curcumin encapsulation. The successful preparation of a series of colloidal carrier systems loaded with curcumin enhanced its water solubility and bioavailability, which proved the feasibility of preparing the delivery system with the pH-driven method [18,19,20,21,22,23,24,25]. However, the pH-driven method has rarely been applied for the encapsulation of other hydrophobic polyphenols. A potential disadvantage of this method is that the encapsulation of polyphenols needs to be carried out under a highly alkaline conditions, which could facilitate the chemical degradation of polyphenols [26]. Peng and co-workers have demonstrated that the pH-driven method can be used to load certain kinds of lipophilic polyphenols such as curcumin and resveratrol but cannot encapsulate quercetin because the degradation rate of quercetin was extremely fast at alkaline pH values [26].

Liposomes are microcosmic phospholipid vesicles with a bilayered membrane structure. The hydrophilic segments of the phospholipids are located on the inside and outside surface of liposomes, while the hydrophobic tail forming lipid bilayers are separated from aqueous phase [27]. Nanoliposome is a nanometric versions of liposome, which can load bioactive substances with different lipophilicities into the various parts of liposomes, such as phospholipid bilayers, hydrophilic core or bilayer interface [28]. Therefore, nanoliposomes are an extremely promising encapsulation technology in the field of nutraceutical products. The superiority of nanoliposome technology is embodied in the protection of sensitive bioactivators, storage stability, excellent loading capacity, improved bioavailability and sustained-release ability [29]. The utilization of high-pressure homogenizer to prepare liposomes can overcome the traditional liposome preparation method that uses detergents or organic solvents that are either undesirable or not allowed ingredients in foods [25].

Naringenin and naringin, with hydrogen donor count as 3 and 8, respectively, whose hydroxyl groups are non-ionized under acidic conditions (Figure 1). When the pH changes to alkaline, they are easy to be negatively charged due to the deprotonation of the hydroxyl group, which extremely enhances their hydrophilic ability. We hypothesized that naringenin and naringin lose their charges due to the protonation of the same group under neutral and acidic conditions, which would reduce their water solubility. Therefore, when the naringenin/naringin alkaline solution and nanoliposome solutions are mixed, the pH is adjusted to an acidic condition, which causes the naringenin or naringin to be embedded into the hydrophobic region within nanoliposomes. In this study, we first determined the stability of naringenin and naringin under alkaline conditions and the change in solubility of naringenin and naringin in the transition from pH 12.0 to neutral or acidic were also measured, which ensured that they can be encapsulated into liposomes by the pH-driven method. Subsequently, nanoliposomes with different concentrations of naringenin or naringin and different contents of lecithin were prepared by microfluidazition and pH-driven methods, and their encapsulation efficiency and maximum loading capacity were determined. The structures of bioactivator-loaded nanoliposomes, particle size and ζ-potentials of nanoliposomes were determined to understand the bioactivator-loaded procedure using the pH-driven method. Finally, the stability of the naringenin-loaded nanoliposomes was studied. This study may lead to develop novel food-grade colloidal delivery systems and apply to introducing naringenin and naringin into foods, supplements, or drugs.

## 2. Materials and Methods

### 2.1. Chemicals

Naringenin (97%) and naringin (95%) powder were purchased from Macklin (Shanghai, China); lecithin from soybean (>70%) and dimethyl sulfoxide (DMSO) were obtained from Aladdin (Shanghai, China). Hydrochloric acid, sodium hydroxide, sodium dihydrogen phosphate, ethanol and other chemicals were all of analytical purity.

### 2.2. Stability of Naringenin and Naringin in Alkaline Condition

Sodium hydroxide solution (6.0 M) was added to 50 mM sodium dihydrogen phosphate solution to prepare a range of buffers with a pH value ranging from 7.0 to 12.0. Then 0.25 mL of naringenin or naringin ethanol solution (2.0 mg/mL) was mixed with buffer solutions (9.75 mL) of different pH values, respectively. These samples with different pH values were stirred using a magnetic stirrer at ambient temperature. At predetermined every 10 min (ended of 60 min), 0.3 mL of mixture was taken out and diluted with phosphate buffer solution (50 mM, pH 7.0) for the determination of naringenin and naringin content with a UV-vis spectrometer (U-T6A, Yipu Instrument Manufacturing Co., Ltd., Shanghai, China) at 288 and 282 nm, respectively.

### 2.3. Effect of pH Shift on the Solubility of Naringenin and Naringin

Naringenin was dissolved in NaH_2_PO_4_ buffer (50 mM, pH 12.0) at 5.0 mg/mL, then the pH values of the solutions were adjusted to 7.0–2.0 with 6.0 M hydrochloric acid. After storage for 24 h at ambient temperature, the suspension was centrifuged, and the supernatant was diluted with DMSO to an appropriate concentration for UV-vis measurement. The determination process for naringin was similar to that described above.

### 2.4. Narigenin-Loaded and Naringin-Loaded Nanoliposome Preparation

Nanoliposomes were produced by the high-pressure homogenization using a microfluidizer. Soybean lecithin (1.2% and 2.5%, *w*/*v*) was suspended in sodium dihydrogen phosphate solution (5 mM, pH 6.0) and stirred for 4 h to ensure complete hydration. Then the soybean lecithin suspension was passed through the microfluidizer 3 times under a working pressure of 121.0 MPa to fabricate nanoliposomes. The nanoliposome solutions were placed in 4 °C before using.

Naringenin was encapsulated into the nanoliposomes by the pH-driven method. Briefly, a range of alkaline naringenin solutions with various naringenin concentration were fabricated by dissolving naringenin in 0.08 M NaOH solution. The alkaline naringenin solutions and nanoliposome solutions were blended in various proportions, and then the mixed systems were rapidly adjusted to pH 7.0, 6.0 and 5.0 by 1.0 M HCl. The final lecithin content within nanoliposome solutions has two levels of 1.0% and 2.0% (*w*/*v*). Meanwhile, the final naringenin concentrations were 0.75, 1.00, 1.25, or 1.50 mg/mL with 1.0% (*w*/*v*) lecithin and the naringenin concentrations were 2.00 or 3.00 mg/mL with 2.0% (*w*/*v*) lecithin, respectively. All the naringenin-loaded nanoliposome samples were equilibrated at room temperature for 24 h.

A series of naringin-loaded nanoliposome (the final naringin concentrations were 1.00, 1.50, or 2.00 mg/mL with 1.0% (*w*/*v*) lecithin and 3.00 mg/mL with 2.0% (*w*/*v*) lecithin, respectively) were prepared only at the final pH of 6.0 according to the above method.

### 2.5. Encapsulation Efficiency and Loading Capacity of Naringenin and Naringin

The encapsulation efficiency (EE) and loading capacity (LC) were measured by a centrifugation method. Briefly, the bioactivator-loaded nanoliposome suspension was centrifuged at 11,000× *g* for 15 min at 4 °C to remove any unencapsulated bioactivator precipitate. Then, the bioactivator-loaded nanoliposome suspensions were centrifuged with ultrafiltration centrifuge tubes (MWCO: 10 kDa) at 11,000× *g* for 30 min and the subnatant was used to determine the concentration of free bioactivator. The concentration of bioactivator in the supernatant for first centrifugation and the subnatant for second centrifugation were measured using a UV-vis spectrophotometer. The EE and LC of the bioactivator-loaded nanoliposomes without free bioactivator were calculated according to the following expressions:(1)$$\mathrm{EE}\;\left(\%\right)=\frac{m_{N}-m_{F}}{m_{I}}\times 100\;$$
(2)$$\;\mathrm{LC}\;\left(\%\right)=\frac{m_{N}-m_{F}}{m_{M}}\times 100\;\;\;\;\;\;\;\;\;$$
where $m_{N}$ is the mass of bioactivator in the nanoliposome solution, $m_{F}$ is the mass of free bioactivator in the nanoliposome solution, $m_{I}$ is the initial mass of bioactivator in the system, and $m_{M}$ is the total mass of the bioactivator-loaded nanoliposomes (bioactivator + lecithin).

### 2.6. Naringenin-Loaded and Naringin-Loaded Nanoliposomes Characterization

A combined dynamic light scattering (DLS)-electrophoresis instrument (ZETASIZER PRO, Malvern Instruments, Worcestershire, U.K.) was used to determine the mean particle size, polydispersity index (PDI) and ζ-potentials of naringenin-loaded and naringin-loaded nanoliposomes at 25 °C. The refractive indexes of the nanoliposomes and the dispersion phase water were set to 1.45 and 1.33, respectively. The nanoliposome suspension was diluted 20 times with buffer solution before measuring to avoid multiple scattering interferences.

### 2.7. Storage Stability

The effect of storage conditions (e.g., temperature and time) on the stability of nanoliposome suspensions were determined to assess their long-term physicochemical stability. The samples (1.0% *w*/*v* lecithin) with naringenin concentrations of 1.00 and 1.25 mg/mL were placed in 4 °C, 25 °C and 37 °C within 31 days. During storage, their encapsulation efficiency, mean particle size and ζ-potentials were recorded according to the same procedure described above at different time intervals.

### 2.8. Atomic Force Microscopy

Atomic force microscopy (AFM) was used for obtaining the microstructure of the blank nanoliposomes and naringenin-loaded nanoliposomes with a concentration of 1.0 and 1.25 mg/mL naringenin before and after 31 days of storage at 4 °C. A droplet of nanoliposome suspension was placed on to a newly cleaved silicon substrate and air-dried for 30 min, and then picture of the naringenin-loaded nanoliposomes were recorded through AFM (C300, Nanosurf, Liestal, Switzerland). AFM images were recorded in tapping mode with a scan time of 3 s and scan points of 512 to obtain sufficient data for statistical analysis.

### 2.9. Statistical Analysis

At least three freshly prepared samples were measured, and the results were expressed as mean ± standard deviation. Analysis of variance (ANOVA) was used to determine significance at the *p* < 0.05 level using Tukey’s HSD test.

## 3. Results and Discussion

### 3.1. Stability of Naringenin and Naringin in Alkaline Conditions

Naringenin molecule contains three hydroxyl groups that can be used as hydrogen donors and their pKa values are 7.86, 9.20 and 9.79 (obtained from chemicalize.com), respectively. The hydroxyl groups could be gradually ionized around and above their pKa value, leading to an increase in the negative charge of the naringenin and its hydrophilicity. Since naringenin was dissolved under alkaline conditions, it is necessary to investigate its stability under alkaline conditions. The chemical stability of naringenin at various pH conditions (from 12.0 to 7.0) was evaluated through measuring the change in the concentration remaining versus time (Figure 2a). The degradation rate of naringenin was about 5% when stored for 10 min at pH 12.0 and the degradation did not increase as the time was extended to 60 min. Ionized hydroxyl groups (phenate ion) of the polyphenols at high pH values are particularly susceptible to oxidation causing the degradation of polyphenols [26]. With the decrease of pH from 11.0 to 7.0, naringenin was relatively stable at this pH range and rarely degraded within 60 min.

Compared with naringenin, naringin molecule contains eight hydroxyl groups that can be used as hydrogen donors, and its solubility in alkaline solutions is higher than naringenin. The change of naringin remaining in alkaline solutions over time is shown in Figure 2b. The degradation rate of naringin was about 8% when stored for 60 min at pH 12.0 and 11.0, suggesting that naringin was relatively stable in alkaline solutions within a short time.

### 3.2. Solubility of Naringenin and Naringin with pH-Shift

In order that naringenin can be moved from the aqueous phase into the hydrophobic domain of the nanoliposome instead of simply blending in the nanoliposome solution, we determined the solubility of naringenin after the pH-shift. Naringenin with a concentration of 5.00 mg/mL could quickly dissolved in a pH 12.0 phosphate buffer solution with a color of orange-yellow (Figure 3a). After adjusting the pH to neutral and acidic, naringenin could quickly restore electrical neutrality, resulting in reduced solubility and precipitated from solution. The solubility of naringenin decreased as the solution of pH-shift range increased. Similar to other hydrophobic polyphenols, such as curcumin, quercetin and resveratrol, the hydroxyl groups on their molecules became electronegative in a strongly alkaline solution which increased their water solubility, but then their water solubility decreased due to the molecules regaining neutrality when the pH was adjusted to neutral [26]. When the naringenin solution changed the pH from 12.0 to 7.0, the solubility of naringenin decreased from 5.00 mg/mL to 94.59 µg/mL, indicating that about 98% of naringenin had been precipitated.

Unlike naringenin, when naringin was dissolved in a NaH_2_PO_4_ buffer of pH 12.0 and then the pH was adjusted to acidity, naringin did not immediately precipitate out but some naringin slowly precipitated over time. However, even after standing for 24 h, the solubility of naringin in the pH 6.0 buffer still reached 592.32 µg/mL, which was about ten times that of naringenin (Figure 3b). Additionally, the solubility of naringin did not decrease as the transition pH decreases (Figure 3b). This result means that when using the pH-driven method to prepare naringin-loaded nanoliposomes, some free naringin may be blended in the nanoliposomes solution.

### 3.3. Encapsulation Efficiency (EE) and Loading Capacity (LC) of Naringenin and Naringin in Nanoliposomes

The EE and LC of nanoliposomes fabricated with different naringenin and lecithin concentrations at pH 7.0, 6.0 and 5.0 were determined, respectively (Figure 4). When the concentration of naringenin was 0.75 mg/mL within 1.0% *w/v* lecithin, the EE of naringenin in the nanoliposomes increased as the shift of pH went down during preparation, being 70.12%, 90.30% and 95.34% for pH 7.0, 6.0 and 5.0, respectively. The same trend appeared at naringenin concentration of 1.00 and 1.25 mg/mL, and the EE of 1.00 mg/mL nanoliposome were 63.89%, 86.49% and 91.47%, while 1.25 mg/mL nanoliposome were 63.47%, 81.39% and 85.20% for pH 7.0, 6.0 and 5.0, respectively. This is may be because more naringenin emerged into the hydrophobic cavity of the liposome as the decreased solubility of naringenin from alkaline solution (pH 12.0) to weakly acidic solution, so the EE of naringenin-loaded nanoliposomes increased [26]. However, when the naringenin concentration reached 1.50 mg/mL, the EE decreased as the pH decreased. When the feeding level of naringenin raised, the EE of nanoliposome gradually decreased, and the nanoliposome solution gradually changed from transparent to cloudy (Figure 4c). The maximum load of naringenin in the nanoliposome solutions (1.0% *w/v* lecithin + 1.25 mg/L naringenin) was 1.02 mg/mL at pH 6.0. When the concentration of naringenin attained 1.50 mg/mL, the maximum LC and EE began to decrease significantly, which means that nanoliposome may began to aggregate and easily formed naringenin crystals to initiate precipitation. Therefore, we tried to increase the amount of lecithin to load more naringenin. When the lecithin content reached 2.0% (*w*/*v*) and the concentration of naringenin was 2.00 mg/mL, the amount of naringenin loaded in nanoliposomes was 1.77 mg at pH 6.0, which was about twice than 1.00 mg/mL with 1.0% (*w*/*v*) lecithin, and their EE were close as 88.56% and 86.49%, respectively. Wang and co-workers prepared naringenin-loaded liposomes by thin-film hydration method, and the EE of the naringenin-loaded liposomes (4.0 mg/mL, 6% *w/v* lecithin) was 72.2% [13]. Compared with the thin-film hydration, the pH-driven is a simple and effective method without the use of high temperature or organic solvents, and also has a higher EE [24,25,26].

LC was also used to evaluate the efficiency of loading naringenin with lecithin. The maximum loading capacity appeared as a naringenin concentration of 1.25 mg/mL were 9.23% and 9.63% for pH 6.0 and 5.0, while it was 8.63% at 1.50 mg/mL naringenin concentration for pH 7.0. The maximum loading capacity of nanoliposomes at pH 6.0 and 5.0 was close (without significant difference) when enough feeding naringenin was provided, which suggesting that the LC of the nanoliposome itself would not be affected in these pHs. When the nanoliposomes were prepared at pH 6.0, the loading capacity increased from 6.34% to 9.23% with the increase of naringenin concentration from 0.75 to 1.25 mg/mL and then dropped to 8.11% for 1.50 mg/mL naringenin.

The EE and LC of nanoliposomes fabricated with different naringin and lecithin concentrations at pH 6.0 were determined, respectively (Figure 5). After removing the free naringin by centrifugation and ultrafiltration, the encapsulation efficiency of naringin-loaded nanoliposomes was between 45.67%–64.54%. Moreover, the free naringin in the naringin-loaded nanoliposome solutions was between 17.78%–39.60%, which was contribute to the low encapsulation efficiency. The loading capacity of naringin-loaded nanoliposomes were 6.06%, 8.01%, 8.37% and 8.24% at naringin concentration of 1.0, 1.5, 2.0 and 3.0 mg/mL, respectively. Compared with naringenin-loaded nanoliposomes, the naringin-loaded nanoliposome solution did not become cloudy as the concentration of naringin increased (Figure 5b). Additionally, the maximum encapsulation efficiency and loading capacity of the naringin-loaded nanoliposomes were lower, which may attribute to its higher molecular weight and solubility in pH 6.0.

Above all, these results suggested that naringenin can be encapsulated into nanoliposomes by an easy pH-driven method according to the reduction in its water-solubility after the pH changed to acidity, which causes the naringenin can be moved from the aqueous solution into the lipid bilayers within nanoliposomes (Figure 5c). However, the naringin-loaded nanoliposomes contained some free naringin in solutions due to its higher water-solubility at acidic conditions (Figure 5d).

### 3.4. Characterization of Naringenin-Loaded and Naringin-Loaded Nanoliposomes

The mean particle size, PDI and ζ-potential changes of blank nanoliposomes and naringenin-loaded nanoliposomes were determined using DLS, as showed in Figure 6. The particle size of the blank nanoliposomes after passing through the microfluidizer consists of 1.0% or 2.0% *w/v* lecithin were about 33 and 35 nm, respectively. The results showed that the influence of different pH values and lecithin concentration on mean particle size changes of blank nanoliposome seems to be negligible (Figure 6a). When the naringenin was loaded into nanoliposomes, the particle size of nanoliposomes increased significantly. For the shift of pH, all the mean particle size of the naringenin-loaded nanoliposomes increased as the pH decreased. When the lecithin content in the system was 1.0% (*w*/*v*), the average diameters of nanoliposomes increased as the concentration of naringenin increased, which was related to the increased encapsulation efficiency and suggested that the larger particle diameter while the more naringenin loaded into nanoliposomes. The average diameters of the nanoliposomes with naringenin concentration of 1.00 mg/mL were 53.47, 58.85 and 62.63 nm for pH 7.0, 6.0 and 5.0, respectively. While the naringenin concentration was 1.25 mg/mL, the average diameters of the nanoliposomes were 61.56, 75.71 and 83.48 nm for pH 7.0, 6.0 and 5.0, respectively. For the 2.0% (*w*/*v*) lecithin concentration, the average diameters of nanoliposomes increased similarly as the shift of pH decreased and the concentration of naringenin increased. Particularly, the average diameters of nanoliposomes with a naringenin concentration of 2.00 mg/mL were 52.20, 57.68 and 63.32 nm for pH 7.0, 6.0 and 5.0, and with a naringenin concentration of 3.00 mg/mL were 75.44, 99.40 and 104.40 nm for pH 7.0, 6.0 and 5.0, respectively. The PDI of all the nanoliposome samples was relatively small (PDI < 0.3) apart from the 0.75 mg/mL naringenin-loaded nanoliposomes, suggesting that their particle size distributions were relatively narrow (Figure 6b,c). Compared with the blank nanoliposome, the PDI of the naringenin-loaded nanoliposomes with a naringenin concentration of 0.75 mg/mL increased, indicating that part of the unloaded and loaded nanoliposomes form a fairly wide particle size distribution (Figure 6c). When the feeding concentration of naringenin increased, the PDI of nanoliposomes decreased slightly. In particular, for the concentrations of 1.50 and 3.00 mg/mL naringenin, the PDI of nanoliposomes dropped significantly, from 0.262 (blank nanoliposome) to 0.164 and 0.149, respectively. The reason may be that the hydrophobicity of nanoliposomes increased after loading naringenin, which inhibited the aggregation of blank nanoliposomes, and the number of blank liposomes should be very small under this condition [22,30]. The ζ-potentials of those samples were negative and ranged from –14.01 to −19.30 mV (Figure 6d), and the pH and concentration of naringenin didn’t seem to cause significant changes in the ζ-potentials of nanoliposomes.

Different from the nanoliposomes loaded with naringenin, the particle size, PDI and ζ-potentials of the naringin-loaded nanoliposomes did not change significantly after loading with naringin (Table 1). Unlike naringenin, naringin may has weak hydrophobicity and electrostatic interaction with nanoliposomes without forming a larger particle size [30].

### 3.5. Stability of Naringenin-Loaded Nanoliposomes

Liposomes need to have good storage stability in order to become a commercial product, which means that they must remain integrity during the entire life cycle of the product. We first investigated the change of the encapsulation efficiency of 1.00 and 1.25 mg/mL naringenin loaded nanoliposomes at 4 °C, 25 °C and 37 °C (Figure 7b–d). At 4 °C, the encapsulation efficiency of 1.00 mg/mL naringenin-loaded nanoliposomes solution remained constant at about 86% after one month of storage, while the nanoliposome solution with a concentration of 1.25 mg/mL dropped from 83.58% to 70.48%. At 25 °C, the encapsulation efficiency of all the nanoliposome solutions decreased, to 74.67% and 50.24%, respectively. At 37 °C, the encapsulation efficiency of 1.00 mg/mL naringenin loaded nanoliposomes solution decreased from 86.23% to 69.69%, and the 1.25 mg/mL naringenin loaded nanoliposomes solution decreased from 83.58% to 54.26%. With the increase of storage temperature, the nanoliposome solution finally changed from milky white to yellow at higher temperature (Figure 7a), indicating that the reduction of encapsulation efficiency is related to the precipitation of naringenin. These results suggest that at high temperatures, nanoliposomes loaded with more naringenin are more hydrophobicity and likely to accelerate aggregation, oxidation, leakage and other chemical reactions during storage, which cause the precipitation of naringenin crystals and the reduction in encapsulation efficiency [22]. Similar to previous research results, conventional liposomes without modification are thermodynamically unstable systems that tend to aggregate, fuse, degrade, or hydrolyze, causing leakage of loaded compounds [31,32,33]. However, the combination of synthetic polymers or biopolymer with liposomes can easily improve the stability and site-specific targeting of liposomes through surface modification [33,34,35].

The storage stability of the nanoliposomes were also determined by recording changes in their mean particle size, PDI and ζ-potentials (Figure 8). There was little change in the mean particle size, PDI and ζ-potentials of the 1.00 and 1.25 mg/mL naringenin loaded nanoliposomes solutions when they were placed in 4 °C within 31 days. At 25 °C, the mean particle size of 1.00 mg/mL naringenin loaded nanoliposomes increased from 65.88 nm to 87.14 nm at 17-day, then slightly decreased to 83.83 nm at day 31. Similarly, the 1.25 mg/mL naringenin loaded nanoliposomes increased from 91.15 nm to 95.80 nm at 10-day, then slightly decreased to 93.55 nm at day 28. The change of the particle size of nanoliposomes with different naringenin concentration at 37 °C also followed the trend of first increasing and then decreasing. These results indicate that storing naringenin-loaded nanoliposomes at high temperature is prone to aggregation in the early stage, and then the particle size decreased in the later stage due to oxidation and leakage of the nanoliposomes [19]. In the early stage of storage, the aggregation of nanoliposomes may cause little increase the PDI, while the PDI of the later supersaturated nanoliposomes may decrease due to the precipitation of part of naringenin. Similar to Homayoonfal et al.’s study of the stability of anthocyanin compound loaded nanoliposomes at 37 °C, there were no significant difference in the parameters of PDI and ζ-potential after one month of storage, but the particle diameter and encapsulation efficiency significantly increased and decreased, respectively [36]. Although the instability mechanism is not yet clear, it may be because of the aggregation and leakage of nanoliposomes and chemical degradation of naringenin in nanoliposomes solutions, which results in color changes or precipitation at high store temperature [20,37,38].

### 3.6. Micosturcture of Naringenin-Loaded Nanoliposomes

The microstructure of the blank nanoliposomes and naringenin loaded nanoliposomes were obtained using AFM. Figure 9 illustrates representative 3D pictures of the blank and nanoliposomes with different naringenin concentration obtained before and after storage at 4 °C for 31 days. Even the nanoliposomes after storage for 31 days, all the nanoliposomes displayed as smooth sphericities that were uniformly distributed throughout the whole pictures, and the particle sizes were similar with those measured by laser granulometry method.

## 4. Conclusions

In conclusion, we have proved that naringenin could be encapsulated into nanoliposomes by an easy pH-driven method according to the reduction in its water-solubility after the pH changed to acidity, which causes the naringenin can be moved from the aqueous solution into the lipid bilayers within nanoliposomes, while the naringin-loaded nanoliposomes contained some free naringin due to its higher water-solubility at lower pH values. Experiments showed that the naringenin-loaded nanoliposomes were predominantly nanometric, negatively charged and exhibited relatively high encapsulation efficiency. Moreover, the naringenin-loaded nanoliposomes still maintain good stability during storage at 4 °C for 31 days, while they were unstable at high store temperatures. In general, an easy method for manufacturing naringenin-loaded nanoliposomes has been constructed that may help to develop novel food-grade colloidal delivery systems and apply to introducing naringenin or other lipophilic polyphenols into foods, supplements, or drugs.

## Figures and Tables

**Figure 1 foods-10-00963-f001:**
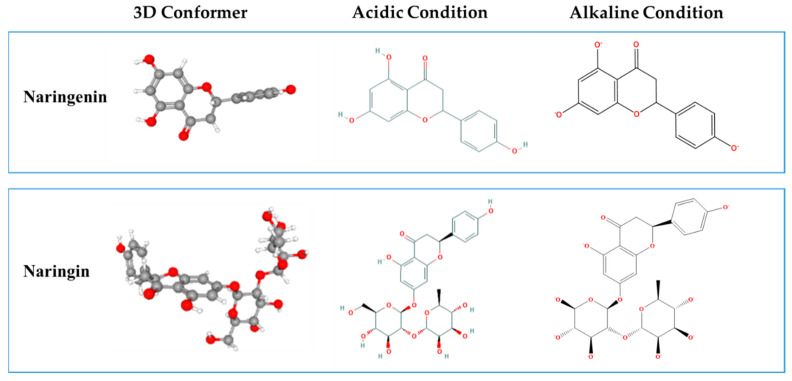
Three dimensions conformer of naringenin and naringin and their hydroxyl groups are non-ionized in acidic condition and negatively charged in alkaline condition.

**Figure 2 foods-10-00963-f002:**
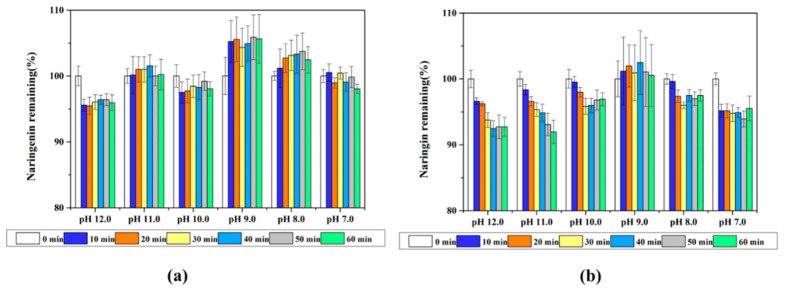
The change of naringenin (**a**) and naringin (**b**) remaining in different alkaline solutions over time.

**Figure 3 foods-10-00963-f003:**
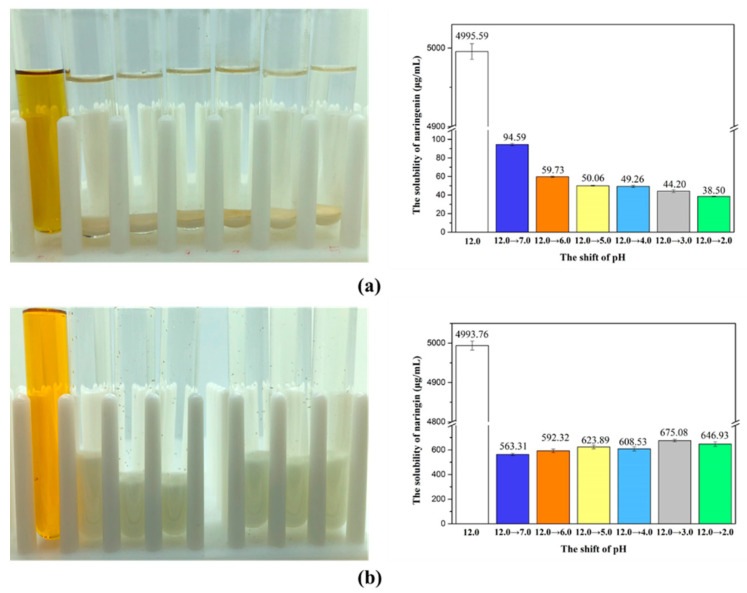
The image and solubility of naringenin (**a**) and naringin (**b**) dissolved at pH 12.0 solution (5.00 mg/mL) and then these samples were adjusted from pH 12.0 to 7.0, 6.0, 5.0, 4.0, 3.0 and 2.0 (left to right), which caused the bioactivators to restore electrical neutrality and reduce solubility.

**Figure 4 foods-10-00963-f004:**
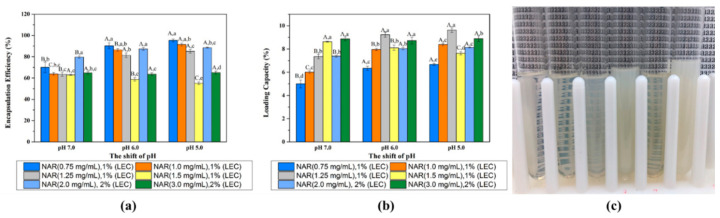
Encapsulation efficiency (**a**) and loading capacity (**b**) of naringenin-loaded nanoliposomes with pH shift to 7.0, 6.0 and 5.0; The image (**c**) of naringenin-loaded nanoliposome solutions with naringenin concentrations of 0.75, 1.00, 1.25 and 1.50 mg/mL with 1% (*w*/*v*) lecithin, while naringenin concentrations of 2.00 and 3.00 mg/mL with 2% (*w*/*v*) lecithin from left to right at pH 6.0. Samples denoted with different letters (A–C) and (a–d) were significantly different (*p* < 0.05) when compared between different pH regions (same naringenin level) and different naringenin levels (same pH region), respectively.

**Figure 5 foods-10-00963-f005:**
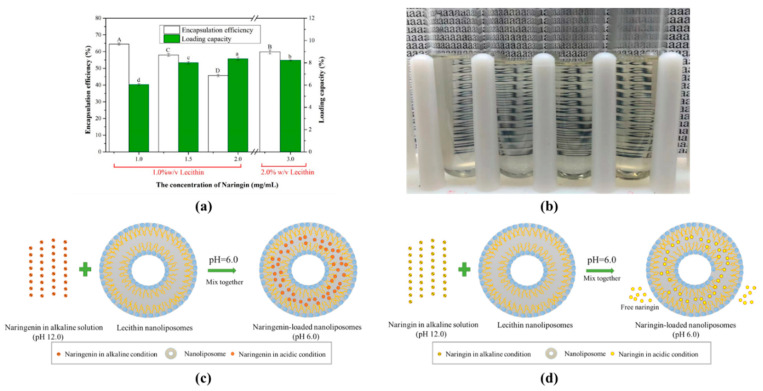
Encapsulation efficiency and loading capacity (**a**) of naringin-loaded nanoliposomes with pH shift to 6.0; The image (**b**) of naringin-loaded nanoliposome solutions with naringin concentrations of 1.0, 1.5, and 2.0 mg/mL with 1% (*w*/*v*) lecithin, while naringin concentrations of 3.0 mg/mL with 2% (*w*/*v*) lecithin from left to right at pH 6.0. Schematic mechanism of the formation of naringenin-loaded (**c**) and naringin-loaded (**d**) nanoliposomes based on pH-driven method. Samples denoted with different letters (A–D) and (a–d) were significantly different (*p* < 0.05) in EE and LC when compared between different naringin levels, respectively.

**Figure 6 foods-10-00963-f006:**
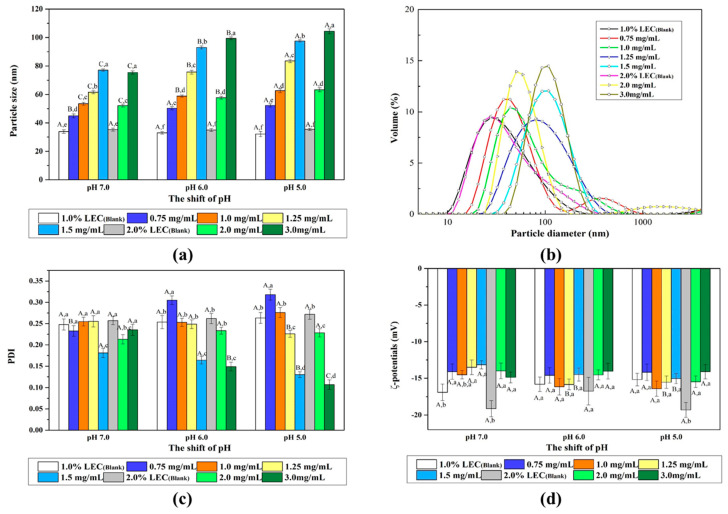
Particle size (**a**), particle size distribution (**b**) at pH 6.0, polydispersity index (**c**, PDI) and ζ-potentials (**d**) of naringenin-loaded nanoliposomes with pH shift to 7.0, 6.0 and 5.0. Samples denoted with different letters (A–C) and (a–f) were significantly different (*p* < 0.05) when compared between different pH regions (same naringenin level) and different naringenin levels (same pH region), respectively.

**Figure 7 foods-10-00963-f007:**
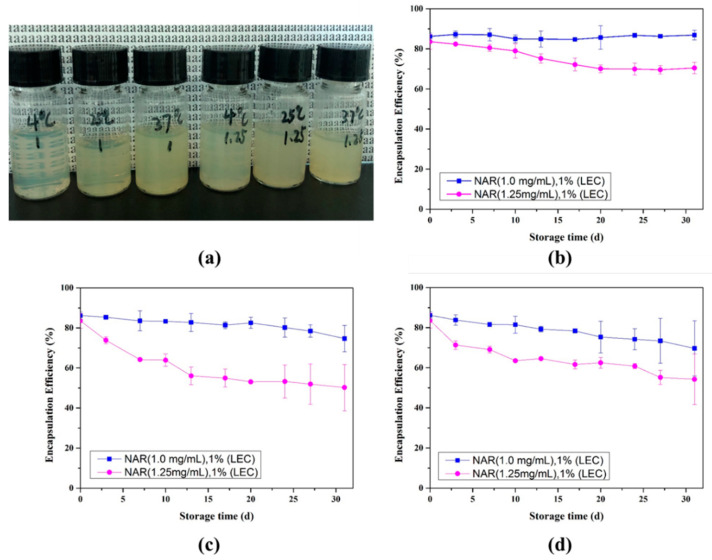
The image (**a**) of naringenin loaded nanoliposome solutions with naringenin concentrations of 1.00 and 1.25 mg/mL with 1% (*w*/*v*) lecithin after 31 days storage at different temperatures; Changes of encapsulation efficiency at 4 °C (**b**), 25 °C (**c**) and 37 °C (**d**).

**Figure 8 foods-10-00963-f008:**
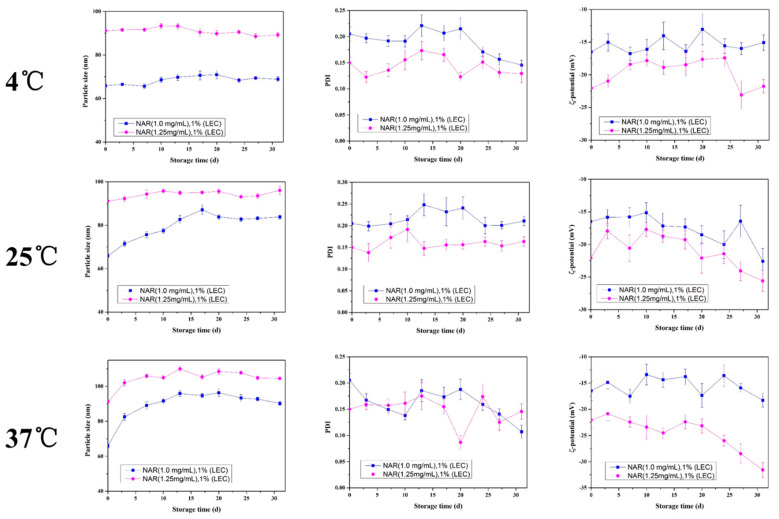
Changes of the naringenin-loaded nanoliposomes in mean particle size, PDI and ζ-potential during storage at 4, 25, and 37 °C for 31 days.

**Figure 9 foods-10-00963-f009:**
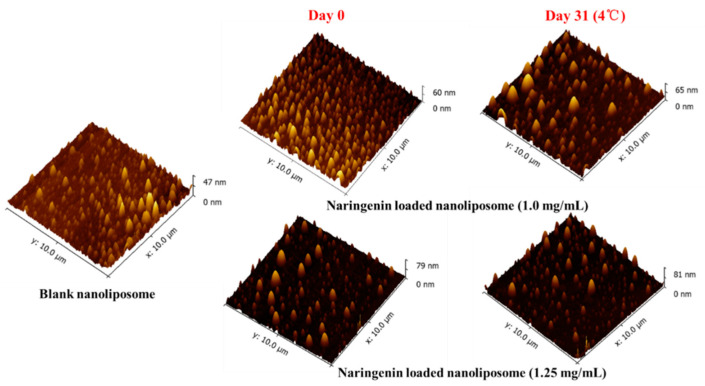
Atomic forces microscopy 3D images of blank nanoliposomes and naringenin-loaded nanoliposomes with a concentration of 1.00 and 1.25 mg/mL after 0 and 31 days of storage at 4 °C.

**Table 1 foods-10-00963-t001:** The mean particle size, PDI and ζ-potentials of naringin-loaded nanoliposomes.

Naringin Concentration (mg/mL)	Mean Particle Size (nm)	PDI	ζ-Potentials (mV)
Blank nanoliposomes	34.93 ± 0.98 ^a^	0.262 ± 0.012 ^b^	−16.74 ± 1.87 ^a^
1.0 (1% *w/v* lecithin)	34.03 ± 0.69 ^a^	0.230 ± 0.010 ^a^	−12.94 ± 1.31 ^a^
1.5 (1% *w/v* lecithin)	34.25 ± 0.72 ^a^	0.237 ± 0.010 ^a,b^	−16.16 ± 1.60 ^a^
2.0 (1% *w/v* lecithin)	35.09 ± 0.86 ^a^	0.240 ± 0.012 ^a,b^	−13.90 ± 1.07 ^a^
3.0 (2% *w/v* lecithin)	35.71 ± 0.49 ^a^	0.260 ± 0.011 ^b^	−16.97 ± 1.48 ^a^

Samples denoted with letters (a,b) were significantly different (*p* < 0.05) when compared between different naringin levels.

## Data Availability

Not applicable.

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
