# Peer review of "Encapsulation of Hydrophobic and Low-Soluble Polyphenols into Nanoliposomes by pH-Driven Method: Naringenin and Naringin as Model Compounds"

_foods, 2021, doi:10.3390/foods10050963_

Round 1

Reviewer 1 Report

In the present manuscript, a simple pH-driven method was used to load naringenin or naringin into nanoliposomes based on the decrease in their water-solubility after the pH changes to acidity.

Lines 24-27: correct syntax.

Line 46: correcto to "polyphenols".

Line 82: correct to "become".

Figure 2: use bars instead of lines or present the results in a table.

Figure 3: the control treatment values should be added in the FIgures for comparison purposes.

Section 3.2: add some discussion.

Figure 6c: correct the statistics in the second column (blue) from the left 

Author Response

Dear reviewer,

We provide a point-by-point response to the reviewer’s comments with a vertion of Word. Please see the attachment, thanks.

Reviewer 2 Report

The paper entitled: "Encapsulation of Hydrophobic and Low-soluble Polyphenols into Nanoliposomes by pH-Driven Method: Naringenin and Naringin as Model Compounds" by Chen et al is an interesting paper regarding the formulation of nanoliposomes encapsulating natural products like Narigenin and Narignin.

Major Comments

-The second paragraph of "Introduction" refers to "curcumin", which sounds to be irrelevant to this study.

-Soybean lecithin was used for the liposome formulation. The authors have to further explain why they studied different pH. Is Soybean lecithin neutral at the studied pH conditions? If it is neutral, how do they believe that affects liposomes loading, the different pH conditions? Do the authors need to take under consideration pKa of these two natural products?

- It misses from the manuscript both in Introduction and in Conclusion the soundness of this study. More data is needed as to link these findings with food industry.

- Data is missing regarding naringin- liposomes, stability, loading and storage conditions stability

Minor Comments

line 23: need to be restructured.... " which causes the naringenin or naringin can be driven ..."

line 26: need to be restructured.... " A series of different the shift of pH..."

line 41: "...7th carbon..."

line 110: It is rather unusual to start a new paragraph with number.

Author Response

(The authors gave the same response as above.)

Reviewer 3 Report

The paper presents a well-structured and innovative research question. Neverthless there are some minor errors you should check. 
Line 83: You should better describe the figure 1. In this way it is not clear.
Line 104: maybe you should better describe this paragraph as "chemicals" instead of materials. 
Line 107: "DMSO" as it refers to an acronyrm it would be better to specify first its entire name;
Line 121: "6.0 M"
Line 129: What do you mean by "crude liposome"? You should better describe this point;
Line 149-151: check the way you write "11000 or 11,000 xg". They should be written in the same way.
line 153: there's an error "sencond";
Line 190: "P" has to be written in a lowercase letter.
Line 194: When you speak about "Chemdraw" at this point, it is a little bit messy. You should first describe this software in Materials and Methods
Line 211: there's an error: "alkali";
FIGURES: In general, all figures could be arranged to the format and aligned in the text. Furthermore, the caption of figures n. 1, 2, 3 should be better described. The graphs in figure 3 should be coloured with softer colours.

Generally, a minor English check is required.

Author Response

(The authors gave the same response as above.)
